# Co-Expression Analysis of Airway Epithelial Transcriptome in Asthma Patients with Eosinophilic vs. Non-Eosinophilic Airway Infiltration

**DOI:** 10.3390/ijms24043789

**Published:** 2023-02-14

**Authors:** Pawel Kozlik-Siwiec, Sylwia Buregwa-Czuma, Izabela Zawlik, Sylwia Dziedzina, Aleksander Myszka, Joanna Zuk-Kuwik, Andzelika Siwiec-Kozlik, Jacek Zarychta, Krzysztof Okon, Lech Zareba, Jerzy Soja, Bogdan Jakiela, Michał Kepski, Jan G. Bazan, Stanislawa Bazan-Socha

**Affiliations:** 1Department of Internal Medicine, Jagiellonian University Medical College, 31-066 Krakow, Poland; 2Haematology Clinical Department, University Hospital, 31-501 Krakow, Poland; 3College of Natural Sciences, Institute of Computer Science, University of Rzeszow, Pigonia 1, 35-310 Rzeszow, Poland; 4Centre for Innovative Research in Medical and Natural Sciences, Institute of Medical Sciences, Medical College, University of Rzeszow, Kopisto 2a, 35-959 Rzeszow, Poland; 5Institute of Medical Sciences, Medical College, University of Rzeszow, Kopisto 2a, 35-959 Rzeszow, Poland; 6Haematology Department, Jagiellonian University Medical College, 31-501 Krakow, Poland; 7Rheumatology and Immunology Clinical Department, University Hospital, 31-688 Krakow, Poland; 8Pulmonary Hospital, 34-736 Zakopane, Poland; 9Department of Pathology, Jagiellonian University Medical College, 33-332 Krakow, Poland

**Keywords:** asthma, eosinophilic, epithelial, bronchial, airway remodeling, differential expression, differential co-expression, basement membrane, transcriptome

## Abstract

Asthma heterogeneity complicates the search for targeted treatment against airway inflammation and remodeling. We sought to investigate relations between eosinophilic inflammation, a phenotypic feature frequent in severe asthma, bronchial epithelial transcriptome, and functional and structural measures of airway remodeling. We compared epithelial gene expression, spirometry, airway cross-sectional geometry (computed tomography), reticular basement membrane thickness (histology), and blood and bronchoalveolar lavage (BAL) cytokines of *n* = 40 moderate to severe eosinophilic (EA) and non-eosinophilic asthma (NEA) patients distinguished by BAL eosinophilia. EA patients showed a similar extent of airway remodeling as NEA but had an increased expression of genes involved in the immune response and inflammation (e.g., *KIR3DS1*), reactive oxygen species generation (*GYS2*, *ATPIF1*), cell activation and proliferation (*ANK3*), cargo transporting (*RAB4B*, *CPLX2*), and tissue remodeling (*FBLN1*, *SOX14*, *GSN*), and a lower expression of genes involved in epithelial integrity (e.g., *GJB1*) and histone acetylation (*SIN3A*). Genes co-expressed in EA were involved in antiviral responses (e.g., *ATP1B1*), cell migration (*EPS8L1*, *STOML3*), cell adhesion (*RAPH1*), epithelial–mesenchymal transition (*ASB3*), and airway hyperreactivity and remodeling (*FBN3*, *RECK*), and several were linked to asthma in genome- (e.g., *MRPL14*, *ASB3*) or epigenome-wide association studies (*CLC*, *GPI*, *SSCRB4*, *STRN4*). Signaling pathways inferred from the co-expression pattern were associated with airway remodeling (e.g., TGF-β/Smad2/3, E2F/Rb, and Wnt/β-catenin).

## 1. Introduction

Asthma is a heterogeneous, chronic inflammatory disease of the airways with a complex pathomechanism and varying clinical presentation. A quarter of asthma patients, particularly those with severe disease, develop progressive and irreversible airway obstruction, leading to persistent symptoms despite optimal treatment. Such a course of the disease is attributed to airway remodeling, i.e., various structural changes associated with epithelial damage, epithelial–mesenchymal transition, subepithelial fibrosis, hyperplasia of airway smooth muscle and mucous glands, and neovascularization, which lead to airway wall thickening, fixed obstruction, and airflow limitation. Current treatment options do not target airway structural changes, and since no accurate biomarker or clinical predictor of ongoing airway remodeling is known, there is no reliable method of indicating those who could benefit from experimental methods, e.g., novel biological therapies before the process becomes clinically symptomatic or apparent in imaging studies [1].

Depending on the type of granulocytes infiltrating the airway mucosa, asthma can be divided into four different inflammatory phenotypes: isolated eosinophilic, isolated neutrophilic, mixed granulocytic, and pauci-granulocytic [2]. Eosinophilic asthma (EA), characterized by the presence of eosinophilic infiltration of the bronchial mucosa and airway secretions, accounts for approximately half of the severe disease cases [3]. This phenotype is characterized by a higher rate of hospital and intensive care unit admissions, more frequent use of oral glucocorticoids, and a higher rate of persistent airflow limitation resulting from airway remodeling driven by T2-profile cytokines and effector cells, particularly eosinophils [4,5]. Finding therapeutic targets behind airway inflammation and remodeling specific to this phenotype presents a vital research goal since available biological therapies against T2 inflammatory responses do not relieve all EA cases [6].

The definition of EA varies between studies, with criteria being based on sputum or bronchoalveolar lavage fluid (BAL) eosinophilia or, more commonly, on increased blood eosinophile count as a surrogate biomarker. No consensus exists as to the cut-off points; those frequently encountered in the literature include sputum eosinophilia of ≥3% [7], BAL eosinophilia of ≥1% [8] or blood eosinophilia ≥ 150–300/mcL [9]. Some studies indicate BAL as the superior reference sample due to the established correlation with clinical asthma features (e.g., severity) or T2-driven inflammatory biomarkers (i.e., the fraction of exhaled nitric oxide (FeNO)) [8]. Furthermore, it has been shown that both blood eosinophilia and FeNO, while correlated, may have unsatisfactory predictive values for airway eosinophilia [10]. Transcriptomic studies faced a similar problem of non-consistency between blood eosinophilia and airway T2-profile gene expression and clinical presentation [11]. Such discordance suggests that transcriptomic studies conducted on the bronchial epithelium need to use EA criteria based on the eosinophilia at the site of sample retrieval, i.e., the lower airways.

Due to the advent of novel therapies, e.g., T2-targeting biologics, the recognition of treatable traits has been proposed as a new paradigm for asthma management. Eosinophilic airway inflammation is considered a distinctive “treatable trait” in severe asthma, and airway eosinophilia appears more stable than neutrophilia over yearly periods of observation [12]. Hence, EA is frequently diagnosed in opposition to less distinctive non-eosinophilic asthma (NEA), regardless of the presence of other granulocytes. As a result, such a clinically recognized EA phenotype is heterogeneous and may include patients with “pure” eosinophilic as well as mixed granulocytic inflammation. Transcriptomic studies focused on a “pure” eosinophilic phenotype [13] are thus unrepresentative of many patients clinically diagnosed with EA. Such studies that focused on a ”pure” eosinophilic phenotype showed an up-regulation of eosinophil marker genes (e.g., *CLC*, *DNASE1L3*), T2-related cytokines (e.g., *IL13*, *IL4*, *IL5*), and genes reflecting an epithelial response to T2-mediated inflammation (e.g., *CLCA1* and *POSTN*) in the airways [14]. The epithelial transcriptome of more heterogenous EA patients, including the mixed granulocytic phenotype, is less known.

In our study, we sought to establish differences in bronchial epithelial transcriptomes between EA and NEA and utilize a systems biology approach in the search for novel molecular mechanisms taking part in the pathogenesis of EA. To this end, we compared mRNA transcriptomes and gene interaction networks of differentially expressed and co-expressed genes in the bronchial brushing samples from moderate to severe EA and NEA patients. Additionally, we made correlations between blood and BAL cytokines and biomarkers of airway inflammation, lung function parameters, high-resolution computed tomography proxies of airway remodeling, and gene expression profile. To distinguish EA patients we used a criterion of eosinophilia in the lower airways defined by BAL cell differential (with ≥1% of eosinophils as the cut-off value), regardless of the presence of other granulocytes. This permitted a more heterogeneous EA study group that included both pure eosinophilic and mixed granulocytic asthma, representative of the clinical occurrences of eosinophilic asthma.

## 2. Results

### 2.1. Clinical Characteristics of the Patients

EA and NEA patients did not differ regarding demographic factors and clinical features, including medication use, asthma severity, lung function, and symptom control, except for body mass index (BMI), which was higher in the NEA group (Table 1). Women comprised 65% and 80% of the EA and NEA samples, respectively, with a median age of over 50 years in both groups. More than half of the cases were diagnosed with atopy, 80% with allergic rhinitis, and nearly 50% with gastroesophageal reflux disease. The number of subjects diagnosed with severe asthma was comparable (*n* = 9 in the EA and *n* = 7 in the NEA group). Five (25%) of the EA and two (10%) of the NEA patients were being treated with oral steroids (chi-squared Pearsons’s test *p* = 0.24). Most of the study sample required persistent use of long-acting β_2_-mimetics (EA 80%, NEA 75%; *p* = 0.41) and/or muscarinic antagonists (EA 5%, NEA 20%; *p* = 0.14).

### 2.2. Functional and Structural Measures of Airway Remodeling Were Similar in EA and NEA

Airflow measured by spirometry, including reversibility test, and lung volumes and capacities in body plethysmography did not differ between EA and NEA patients. On high-resolution computed tomography (HRCT), the EA group had higher wall thickness (mean 1.216 vs. 1.078 mm; *p* = 0.03), airway diameter (mean 5.24 vs. 4.59 mm; *p* = 0.04), and wall area (mean 25.37 vs. 18.7 mm^2^; *p* = 0.03) as measured at RB1 bronchus. However, the wall area ratio (WAR) and the wall thickness ratio (WTR), representing more reliable proxies of airway remodeling [1], did not differ between EA and NEA. RBM thickness was also comparable between EA and NEA patients (median of 6.99 and 5.82 mm, respectively; *p* = 0.47) (Table 2). RBM thickness was not associated with any of the CT airway remodeling measures.

### 2.3. BAL and Blood Eosinophils Were Well Correlated

The median percentage of BAL eosinophils equaled 2.3% in the EA group, as compared to 0% in the NEA group. Instead, there were no differences in the median percentage of BAL neutrophiles (4% in EA vs. 3% in NEA, *p* = 0.211) and lymphocytes (8% vs. 4%, respectively; *p* = 0.052).

The median blood eosinophil count was significantly higher in the EA group (480/mcL) compared to the NEA group (140/mcL, *p* = 0.008), with blood eosinophilia of 353/mcL as an optimal cut-off point differentiating between the two in a logistic regression model. Notably, 35% of patients in the EA group had blood eosinophil counts below this threshold. In addition, BAL and blood eosinophil counts were well correlated (r = 0.58, *p* < 0.001).

### 2.4. Blood and Bronchoalveolar Cytokine Profiles Were Separately Correlated with Eosinophilia in Each Compartment

In Table 3, we present the differences between the serum and BAL immune mediators and the biomarkers of airway remodeling. As shown, both asthma groups did not differ regarding the investigated serum and BAL cytokine levels, except for serum IL-10, which was higher in the EA group. Consequently, serum IL-10 was the only cytokine differing between the cases selected by blood eosinophilia and was higher among those at or exceeding the cut-off value of 350/mL (*p* < 0.05). Blood eosinophilia was positively correlated with the serum concentrations of periostin (r = 0.35, *p* < 0.05), IL-6 (r = 0.49, *p* < 0.05), IL-10 (r = 0.66, *p* < 0.05), IL-12p70 (r = 0.39, *p* < 0.05), and ADAM33 (r = 0.59, *p* < 0.05). No associations were documented between blood eosinophilia and BAL cytokine levels. In turn, the percentage of BAL eosinophils correlated positively with serum concentrations of IL-10 (r = 0.39, *p* < 0.05) and IL-6 (r = 0.36, *p* < 0.05) and negatively with the BAL concentration of IL-6 (r = −0.32, *p* < 0.05). These results highlight discrepancies (lack of cross-correlation) between eosinophilia and cytokine levels measured in different compartments.

Airway remodeling proxies in HRCT were not related to blood or BAL eosinophilia. In turn, RBM correlated positively with BAL periostin (r = 0.48; *p* < 0.05) and, surprisingly, negatively with blood periostin (r = −0.51; *p* < 0.05) and blood IL-23 (r = −0.4; *p* < 0.05).

### 2.5. Differentially Expressed Genes Are Involved in Processes Associated with Asthma Pathogenesis

The expression levels of 32 genes differed significantly between EA and NEA; however, the differences were minor (Table 4). EA patients were characterized by a higher expression of 19 genes involved in various cellular processes, such as plasma membrane organization (e.g., *CPLX2*), cell motility, activation and proliferation (e.g., *ANK3*, *SUN5*), cargo transporting (e.g., *RAB4B*), ciliogenesis (e.g., *GSN*), inflammation (e.g., *GYS2*), and reactive oxygen species generation (e.g., *NDUFAF8*, *ENTR1*, *ATPIF*), and were associated with bronchial hyperreactivity (e.g., *RTN4RL1*, *CNN1*) as well as extracellular matrix composition and airway remodeling (e.g., *FMNL1*, *FBLN1*, *GSN*). Conversely, 13 genes showed higher expression in NEA, including those involved in cell apoptosis (e.g., *GJB1*, *ENTR*), the response to hypoxia (e.g., *SIN3A*), and cellular protein localization (e.g., *DNAJ1*). The relevance of the differentially expressed genes for asthma pathogenesis based on the literature review is presented in Table 4.

### 2.6. Differentially Expressed Genes Correlate with Lung Function and Proxies of Airway Remodeling but Not Reticular Basement Membrane Thickness

Expression levels of 11 differentially expressed genes correlated significantly with either FEV_1_ or FEV_1_/FVC (Table 5). For example, both FEV_1_ and FEV_1_/FVC remained in negative associations with four of the genes up-regulated in EA (*GSN*, *NDUFAF8*, *KIR3DS1*, and *KRTAP10.1*) and in positive relationships with one gene down-regulated in EA (*TBC1D12*).

*ATPIF1* correlated positively with the proxies of airway remodeling differing between EA and NEA cases, while *RAB4B*, *GSN*, and *NDUFAF8* were positively correlated with WAR.

Our study sample had no significant associations between RBM thickness and the gene expression profile.

### 2.7. Differential Co-Expression Analysis Reveals Genes Co-Regulated in Eosinophilic Asthma

Hierarchical cluster analysis comparing EA and NEA revealed 23 groups of differentially co-expressed genes. A single group consisting of 32 genes with the highest mean pairwise correlation differences was chosen for subsequent analyses. Table 6 shows a literature-based assessment of individual differentially co-expressed genes for their mechanistic role in EA and bronchial remodeling.

The differentially co-expressed genes were involved in the airway response to viral infections (e.g., *ATP1B1*, *EPS15*), arachidonic acid metabolism (e.g., *FADS6*), cell migration (e.g., *EPS8L1*, *STOML3*, *RHOBTB2*), surface receptors endocytosis (e.g., *STRN4*, *EPS15*, *ATP1B1*) or surface receptor expression (e.g., *CCT7*), oxidative stress response (e.g., *DIO3*, *RHOBTB2*), impaired adhesion (e.g., *ATP1B1*, *RAPH1*, *STOML3*), epithelial–mesenchymal transition (e.g., *ASB3*, *RADX*, *CCT7*, *MRPL14*, *PPP2R3B*, and *SLC19A1*), myofibroblast differentiation (e.g., *CCT7*), smooth muscle proliferation (e.g., *ASB3*, *ATP1B1*), airway hyperreactivity (e.g., *RECK*, *STOML3*, *ATP1B1*, *OR52I1*), extracellular matrix remodeling (e.g., *FBN*, *RECK*), angiogenesis (e.g., *GPI*, *RHOBTB2*), and neuronal pathogenesis of asthma (e.g., *OR52I1*, *STRN4*, *TTC3P1*, *GPI*, *CABP5*).

Moreover, the differentially co-expressed genes, both up- and down-regulated in EA, were previously linked to asthma in the genome- (e.g., *MRPL14* [31], *ASB3* [23], *RHOBTB2* [23]) and epigenome-wide (e.g., *CLC* [23], *EPS15* [32], *GPI* [36], *SRCRB4D* [23], *STRN4* [23]) association studies.

Additional detailed descriptions of molecular functions and biological processes associated with differentially co-expressed genes by gene ontology are listed in Appendix A. They include phenomena significant in the pathogenesis of asthma and airway remodeling, such as T-cell receptor binding, MHC II protein complex binding, immunoglobulin secretion, myeloid cell apoptosis, regulation of cellular response to growth factors (e.g., TGF-β), and metalloendopeptidase inhibitory activity.

Lung-specific protein–protein interactions of the differentially co-expressed genes are displayed in Figure 1. The *CCT7* gene, occupying a central position in the graph, takes part in fibrotic tissue remodeling, including fibrotic wound healing. In contrast, the interacting proteins with the highest connectedness in the graph were related to asthma pathogenesis and include fusogenic protein genes (*ICAM1*, *VCAM1*, *ITGA4*, and their regulator, *ELAVL1*) as well as *ILK*, a β_1_ integrin-linked kinase involved in fibroblast migration, myofibroblast differentiation, and tissue remodeling [37].

### 2.8. Regulatory Networks of the Differentially Co-Expressed Genes and the Inferred Signaling Pathways

Figure 2 shows a regulatory network of the differentially co-expressed genes, indicating their relation to asthma-relevant transcription factors: E2F1, SP1, YY1, MYC, FOS, EGR1, CTCF, and NFKB1 as well as microRNAs: miR-135b [38], miR-135a [39], miR-340 [30], and miR-223 [40], recently described as involved in asthma pathogenesis through, e.g., regulation of CXCL12, signaling via JAK/STAT pathway, expression of MIDI1, and subsequent mTOR signaling or release of proinflammatory CCL20 from bronchial epithelial cells.

Additional regulatory networks of the differentially co-expressed genes are presented in Appendix A. Furthermore, the associated transcription factors were included in a single graph, and their network-disrupting potential expressed as betweenness centrality was plotted against literature coverage for asthma to identify key nodes with low literature representation and thus presenting new research opportunities (Figure 3). Enrichment analysis of the transcription factors revealed multiple associated signaling pathways listed in Appendix A, including SMAD2/3, activator protein-1, E2F/Rb, and Wnt/β-catenin pathways.

### 2.9. Kinase Perturbation Studies

Kinase perturbation studies included in Gene Expression Omnibus (GEO) database indicate differential expression of several differentially co-expressed genes upon knockout of the following kinases (Appendix A):TGF-β receptor II (*TGFBR2*): down-regulation of *CCT7*, *EPS15*, *MRPL14* [41];Homeodomain-interacting protein kinase 1 (*HIPK1*): down-regulation of *GPI*, *MAEA*, *STRN4* [42];Inhibitor of nuclear factor kappa-B kinase subunit epsilon (*IKBKE*): up-regulation of *DGLUCY*, *RAPH1*, *ASB3* [43].

### 2.10. Cell-Type-Specific Histone Modifications Related to the Expression of Differentially Co-Expressed Genes

A considerable proportion of the differentially co-expressed genes (15 out of 32, all up-regulated in the EA) matched differential expression patterns resulting from histone modifications H3K9ac and H3K27me3 characterized for human lung fibroblast hg19 [44] (Appendix A). 

## 3. Discussion

In the present study, we showed that the two samples of consecutive moderate to severe asthma patients, comparable regarding demography, medical history, and disease severity, but diagnosed as having either EA or NEA, differed in the epithelial transcriptome despite shared serum and BAL cytokine profiles, lung function, and measures of airway remodeling. Since our comparison included EA patients with both “pure” eosinophilic as well as mixed granulocytic infiltration against NEA patients with both pauci-granulocytic and neutrophilic phenotypes, the resulting differences are more representative of heterogenous EA patients encountered in clinical practice, but simultaneously less distinctive than in a comparison between “pure eosinophilic” and “pauci-granulocytic” or ”pure neutrophilic” asthma cases. 

While BAL and blood eosinophilia are correlated, which permits the use of the latter as a feasible predictor of response to eosinophil-depleting biologics, tissue eosinophils seem more resistant to therapeutic intervention, and airway eosinophilia can remain despite the resolution of blood eosinophilia. This is concordant with the observed high BAL eosinophilia despite low blood eosinophilia in over a third of our EA patients. As per previous studies, airway rather than blood eosinophilia could be a better discriminator of eosinophilic asthma, especially in patients already treated with systemic steroids.

A similar phenomenon seems to affect T2-profile cytokines in different body compartments (i.e., blood and bronchi), with the surprising negative correlation of RMB thickness with serum (but not BAL) levels of periostin and IL-23 in our study, an unexpected finding given the role of periostin as a biomarker of eosinophilic airway inflammation [45] and remodeling and the role of IL-23 as an enhancer of T2-mediated eosinophilic inflammation, crucial for the maintenance of eosinophil-recruiting Th17 cells [46]. These findings, however, are consistent with earlier studies reporting a correlation between sputum (but not serum) periostin and sputum eosinophilia [47].

While most of the measured blood and BAL biomarkers did not discriminate between EA and NEA cases, epithelial transcriptomes revealed differential expression and co-expression of multiple genes empirically or speculatively associated with asthma pathogenesis.

EA was characterized by the up-regulation of several genes involved in cell activation, proliferation, motility, cargo transporting, immune response, airway hyperresponsiveness, and remodeling. For example, the product of *RAB4B* is co-expressed with MHC II class genes and is thought to enhance antigen presentation [48]. *FMNL1* modifies macrophage motility and is implicated as a major regulatory gene of mild/moderate persistent asthma in children and adults [49]. The *GYS2* gene is up-regulated upon signaling through PPARα, an important regulator of airway inflammation. The product of *SDCCAG3* is necessary for the expression [50] of tumor necrosis factor-α receptor (*TNFR*) on the cell surface, thus contributing to TNF-α-mediated airway hyperresponsiveness [51]. *RN4RL1*, found on vagal sensory neurons, likely contributes to airway hyperreactivity and neuronal hypothesis of asthma pathogenesis [52], similar to *TBC1D12*, the down-regulation of which may result in airway neurite sprouting [53], another feature of airway remodeling. Fibrillin-1 (*FBLN1*) is implicated in stabilizing ECM proteins, including periostin, through regulation of the biological availability of latent TGF-β [16]. In turn, signaling through TGF-β results in the expression of *CNN1* [54], involved in ECM remodeling and airway hyperreactivity [55], as well as *GSN* [17] (gelsolin) and *TMEFF* implicated in epithelial–mesenchymal transition [56]. *ATPIF1* promotes transcriptional activation of *NFKB*, resulting in a proliferative response and tissue remodeling [57].

Accordingly, expression levels of the genes, the function of which aids bronchial remodeling, exhibited positive correlations with proxies of airway remodeling (e.g., *ATPIF1*, *GSN*, *NDUFAF8*, *RAB4B*) and consistently negative correlation with lung function parameters (e.g., *GSN*, *NDUFAF8*, *KIR3DS1*), contrary to negatively correlated *TBC1D12*, the expression of which may prevent unfavorable innervation of airway smooth muscle cells resulting in hypercontractility.

Despite their pathogenetic relevance, the differences in expression levels of the abovementioned genes were low, likely owing to similar clinical characteristics of both asthma patient groups. This remained true of the 32 differentially co-expressed genes; however, the low fold-change ratio, in this case, may indicate a better match in differential co-expression. Furthermore, their expression levels were better correlated within the EA than the NEA group, likely showing higher functional consistency of the EA sample.

Among the differentially co-expressed genes, eight were previously linked to asthma in genome- (*MRPL14*, *ASB3*, *RHOBTB2*) and epigenome-wide (*CLC*, *EPS15*, *GPI*, *SSCRB4*, *STRN4*) association studies and five were mentioned in the literature in the context of asthma (*SLC19A1*, *MAEA*, *CLC*, *ATP1B1*, and *RECK*). For example, *ATP1B1* was previously included in transcriptomic cluster TAC3 of patients with moderate to high sputum eosinophilia in the U-BIOPRED cohort of T2-high asthma, playing a role in the epithelial sheathing, mucus secretion, and airway hyperreactivity, whereas *RECK* is a negative regulator of matrix metalloproteinase-9 (MMP-9), a significant MMP involved in airway remodeling.

Multiple genes co-expressed in EA could be associated mechanistically with different processes underlying asthma and airway remodeling, but so far, have not been described in the context of this disease. For example, *RHOBTB2* is needed for the expression of *CXCL14*, an autocrine growth chemokine for fibroblasts and a chemoattractant controlling dendritic cell activation, leukocyte migration, and angiogenesis. *EPS15*, *CCT7*, *SRPRB*, and *STRN4* are involved in receptor endocytosis and the turnover of β_2_-adrenergic and M_3_-muscarinic receptors critical for bronchial constriction and airway hyperresponsiveness as well as signaling through TGF-β and β_1_-integrins linked to eosinophil recruitment [58] and airway wall structural changes. Affinity capture studies indicate interactions between the protein products of *RPS13*, *CCT7*, and *EPS15* and proteins involved in cell–cell and cell-ECM adhesion molecules relevant for asthma, such as VCAM-1, α_4_β_1_ integrin, and fibronectin. The protein product of *ASB3*, via MAP kinase and Akt phosphorylation, regulates Erk1/2 and PI3K/Akt signal transduction pathways, being implicated in smooth muscle proliferation in asthma.

Several of the differentially co-expressed genes could be linked to Th17 differentiation. For example, *STRN4*, inhibiting MAP4K4 kinase, is considered the critical inhibitor of the Hippo pathway [59] and promotor of Th17 differentiation [60], resulting in IL-6 and IL-17 overproduction. Co-expression of *CLC* and *PSG2*, the two genes involved in the limitation of Th2- and induction of Th17-differentiation in EA patients, suggests both a counterweight mechanism to limit T2-response and a possible role behind the heterogeneity of T2-high asthma, recently proposed to be divided into IL-5-high/IL-17F-high asthma (with mixed granulocytic infiltration) and IL-4/IL-13-high asthma (with eosinophilic infiltration alone).

Since differential co-expression is thought to reveal a common regulation responsible for coordinated gene expression, the activity of signaling pathways, transcription factors, and histone modifications, which can be inferred through gene set enrichment analysis, was reviewed for associations with asthma pathogenesis. This systems biology approach allowed us to place 29 of the 32 differentially co-expressed genes in gene regulatory networks with transcription factors and miRNAs previously described as involved in asthma pathogenesis, indicating highly connected network disruptors of potential therapeutic applications (Figure 2, Appendix A). Topological robustness analysis of the regulatory network of the differentially co-expressed genes identifies highly connected transcription factors with network-disrupting potential, expressed as betweenness centrality. Plotted against the literature coverage for asthma (Figure 3), the analysis reveals understudied transcription factors of possible relevance. For example, AR (androgen receptor) could be regarded as an understudied, highly connected regulator, which was only recently correlated with asthma control [61].

Data from kinase perturbation studies show that several of the differentially co-expressed genes are dependent on the activation of *TGFBR2*, *HIPK1*, and *IKBKE* (Appendix A). The ligand of TGFBR2, TGF-β, has a well-established role in asthma and airway remodeling, and its receptor was found to be involved in T-cell differentiation. The role of HIPK1 has to date only been described in fetal angiogenesis activated by the TGF-β-TAK1 pathway [42], apoptosis induced by TNF-α, and the prevention of MAP3K5-JNK activation. Our study links HIPK1 with eosinophilic asthma through the pattern of differentially co-expressed genes, a suggestion reinforced by the association between MAP3K5 and atopy [62]. IKBKE is a known target of the NFκB, a transcription factor involved in the inflammatory response of asthma pathogenesis and airway remodeling.

Several other nuclear signaling pathways were especially highly enriched for the transcription factors associated with the genes differentially co-expressed in our data (Appendix A). Of those, SMAD2/3 corresponds to the profibrotic effects of the TGF-β/Smad2/3 pathway involved in fibroblast to myofibroblast transition [63], and the E2F/Rb and Wnt/β-catenin pathways were both implicated in airway smooth muscle hypertrophy [64]. This concordance with the current literature seems to validate our results and indicates other less studied pathways (such as Notch-mediated HES/HEY network and HIF-1-α transcription factor network) as possibly relevant for future studies.

It is worth mentioning that 15 differentially co-expressed genes in our results have been up-regulated in human lung fibroblast resulting from histone H3K9ac and H3K27me3 epigenetic modifications. While the latter histone has a pluripotency function, the H3K9ac was shown as being linked to epithelial–mesenchymal transition and extracellular matrix degradation upon treatment of the cell culture with TNF-α and TGF-β [65]. Therefore, up-regulation of the associated genes in our study might suggest the same epigenetic modification leading to ongoing epithelial–mesenchymal transition in EA, consistent with the abundance of eosinophil-derived TGF-β in this disease phenotype. 

Since the designations of eosinophilic and neutrophilic asthma are not mutually exclusive, our sets of differentially expressed and co-expressed genes likely indicate distinctions between broadly defined EA (including mixed granulocytic phenotype) and NEA (including neutrophilic and pauci-granulocytic phenotypes). EA patients showed higher pathogenetic consistency with up-regulation of the genes directly or indirectly involved in T2-dependent inflammation, Th17 polarization, and airway remodeling.

## 4. Materials and Methods

### 4.1. Characteristics of the Patients and Study Design

The study group consisted of 40 asthma patients aged 20–70 years with at least 10-year history of the disease. The diagnosis of asthma was assessed according to the Global Initiative for Asthma (GINA) guidelines [66]. All patients were current non-smokers for at least 5 years prior to enrollment. Except for biological treatment, all asthma medications were permitted, with oral corticosteroids at a daily dose ≤10 mg of prednisolone or equivalent if the dose was unchanged during the preceding 3 months. 

Spirometry with bronchial reversibility test using 400 µg of albuterol and body plethysmography after bronchodilator with an assessment of the residual volume (RV) and total lung capacity (TLC) were carried out in all patients using a Jaeger MasterLab spirometer (Jaeger-Toennies GmbH; Hochberg, Germany). Fasting blood samples were collected for routine laboratory tests and further cytokine measurements. 

Bronchoscopy with bronchial brushings and bronchoalveolar lavage (BAL) fluid sampling was performed in all study participants. Half of the subjects (*n* = 20) were diagnosed with EA defined as having at least 1% of eosinophils in BAL based on cell differential count. The remaining individuals were assigned as NEA (*n* = 20). BAL eosinophilia was chosen as the discriminatory feature due to: (1) the use of airway eosinophilia rather than peripheral blood eosinophilia as the criterion in the U-BIOPRED severe asthma cohort study [67], and (2) the finding that markers associated with T2-inflammation, such as FeNO, blood eosinophilia, and serum periostin, predicted 1% BAL eosinophilia better than 3% sputum eosinophilia [8]. 

The study was approved by the Ethics Committee of the Jagiellonian University under approval number KBET/151/B/2013 and conducted in accordance with the Declaration of Helsinki. All participants gave written informed consent to participate in the study.

### 4.2. Airway Cross-Sectional Geometry in Lung Computed Tomography

Lung CT indices of bronchial remodeling were measured in CT scans performed 1 h after administration of 400 µg albuterol using 64-raw multidetector computed tomography (Aquilion TSX-101A, Toshiba Medical Systems Corporation, Otawara, Japan) in a helical scanning mode. The automated program AW Server (General Electric Healthcare, Wauwatosa, WI, USA) was used to quantify the airway cross-sectional geometry at the right upper lobe apical segmental bronchus (RB1). We measured lumen and wall area, average wall thickness, wall area ratio (WAR, i.e., average difference between the outer and inner areas divided by the outer area), and wall thickness ratio (WTR, i.e., wall thickness divided by the outer diameter). The low-attenuation lung area (LAA%) was employed to quantify the presence of emphysema and/or lung hyperinflation and was calculated automatically using the Volume Viewer 11.3 software (General Electric Healthcare) with 1 mm soft tissue reconstruction algorithm. More details on the measures used were provided in our previous publication [1].

### 4.3. Bronchofiberoscopy and Airway Sampling

Bronchofiberoscopy was performed according to the guidelines of the American Thoracic Society [68] using the bronchofiberoscope BF 1T180 (Olympus, Japan) with local anesthesia (2% lidocaine spray) and in mild sedation (0.05–0.1 mg fentanyl and 2.5–5 mg midazolam, both intravenous). 

Endobronchial forceps biopsies were taken from the right lower lobe (the carina between B9 and B10) together with the bronchial brush biopsies (Boston Scientific, Marlborough, MA). The endobronchial specimens were immediately fixed in 10% neutral buffered formalin solution (Sigma-Aldrich, Saint Luis, MO, USA) and sent to the Pathology Department for further analysis. Brushes were immediately immersed in TRIzol lysis reagent to minimize RNA degradation and kept for further analysis (Thermo Fisher Scientific, Carlsbad, CA, USA). 

BAL samples were centrifuged at 2000× *g* for 20 min. The supernatant was frozen in aliquots and stored at −70 °C until analysis.

### 4.4. Bronchial Biopsy Histology and Reticular Basement Membrane Measurement

Endobronchial biopsy tissue specimens were processed routinely, as described in our previous publications [69]. The 2 µm paraffin-embedded sections were cut and stained with hematoxylin and eosin. The slides were photographed by a Nikon D5300 camera attached to the Zeiss Axioscope microscope with a 100× oil immersion lens. The images were analyzed by the AnalySIS 3.2 software (Soft Imaging System GmbH, Muenster, Germany). The RBM thickness was measured along the biopsy’s epithelial surface according to the orthogonal intercept method suggested by Ferrando et al. [70] using arbitrary distance units. For each patient, at least 30 individual RBM measurements were evaluated at intervals of 9.5 µm. The results were expressed as harmonic mean, defined in our previous publication.

### 4.5. Measurements of Immune Mediators in Serum Samples and BAL Supernatant 

Immunoenzymatic assay (ELISA) was used to measure serum and BAL concentrations of interleukin (IL)-4, IL-5, IL-6, IL-10, IL-12p70, IL-17A, IL-23, and interferon gamma (IFN-γ) (eBiosciencea, Vienna, Austria), periostin (Phoenix Pharmaceuticals, Burlingame, CA, USA), and serum disintegrin and metalloproteinase domain-containing protein 33 (ADAM 33) (Cloud-Clone Corp., Katy, TX, USA).

### 4.6. RNA Isolation and Microarray Processing

Total RNA was extracted from bronchial brush biopsies with mRNA isolation kits (DNAGdańsk, Gdańsk, Poland), fractioned in gravity gradient, isolated in chromatographic columns, and stored at −80 °C. The quality of each sample was assessed using Qiagen^®^ QIAxel, and RNA integrity was verified by agarose gel electrophoresis. The resulting RNA was reverse transcribed into cDNA library using Syngen^®^ UniversalScript Reverse Transcriptase. The product was purified by Syngen ^®^ PCR ME Mini Kit and fluorescently labeled and purified using Kreatech ^®^ ULS Platinum Bright Red/Orange Kit. Hybridization to microarrays occurred on the Human Genomic 49K Mi ReadyArray (a Human Exonic Evidence-Based Oligonucleotide array (HEEBO); Microarray Inc., Huntsville, AL, USA) at 37 °C for 24 h.

### 4.7. Retrieval of Microarray Data

Microarrays were scanned with a InnoScan 900 Microarray Scanner, and hybridization signals were detected using Mapix software (v.6.0.1; Innopsys, Carbonne, France). The software did not perform microarray gridding correctly in some cases. Custom gridding algorithm was developed to avoid grid displacement as previously suggested [71]. The coordinates of positioning markers to establish sub-grids were found using a maximally stable extremal regions (MSER)-based method. Computed results were verified manually and corrected if needed, with overall gridding performance exceeding that of the commercial software, ultimately resulting in expression profiles of 33,519 gene products. Microarray data preprocessing (background correction and normalization) was performed by limma R software using the established methods. After background correction, quantile normalization and filtration for coefficients of gene expression variation between 0.3 and 10, 14,823 annotated gene products were included in subsequent analyses.

### 4.8. Basic Statistical Analysis

Basic statistical analysis was performed using Statistica TIBCO 13.3 and R (version 3.6.1) software. Genomic data were analyzed with Bioconductor v.3.7. software of the R environment [72]. Data distribution normality was verified using the Shapiro–Wilk test. Continuous variables, mostly non-normally distributed, were reported as a median with interquartile range or as a mean with standard deviation, if appropriate, and compared using Mann–Whitney U-test or unpaired *t*-test, respectively. Categorical variables were given as percentages and compared using χ2 test.

### 4.9. Statistical Analysis of Differential Gene Expression and Co-Expression

Two groups of *n* = 20 patients were calculated as sufficient to analyze the total of 14,823 gene products at desired fold differences of 2, desired power of 0.8, a standard deviation of 0.7, and the acceptable number of false positives of 4 using FDR-based sample size calculator [73]. Since differential expression can confound differential co-expression analysis [74], a list of differentially expressed genes was established using ANCOVA to adjust for age, sex, BMI, and oral corticosteroid use. Adjustment for false discovery rate (FDR) was made using Benjamini–Hochberg procedure.

A list of differentially co-expressed genes was established using CoXpress [75] R package. In this method, groups of genes, the expression levels of which are highly correlated in one set of experiments (i.e., EA) but not significantly correlated in the second set of experiments (i.e., NEA), were found. Identified groups of co-expressed genes are thought to be indicative of the underlying coordinated regulation, e.g., signaling pathways or histone modifications specific to the studied condition. A group of genes, the distribution of the pairwise correlations of which was found to be random in NEA (*p* ≥ 0.05) and non-random in EA (*p* < 0.05), was recognized as differentially co-expressed.

### 4.10. Gene Set Enrichment, Interaction Networks, and Their Topological Robustness Analysis

We utilized the systems biology approach to (1) identify genes with coordinated expression in the airways of EA and (NEA) patients (differentially co-expressed genes), (2) establish networks of their binary molecular interactions and regulation using curated resources, and (3) conduct a systematic review of the relevant literature to infer their possible role in the disease pathogenesis in search for novel regulators, biomarkers, and mechanisms of airway inflammation and remodeling.

Associations between differentially co-expressed genes, airway inflammation, and remodeling were investigated using the following workflow: Gene set enrichment analysis (GSEA) to establish histone modifications and cell types matching differentially co-expressed genes’ expression pattern and associated biological processes, molecular functions, and signaling pathways, using Enrichr tool [76] and curated datasets (Appendix A);Analysis of the networks of associated molecular entity interactions, i.e., (1) lung-specific binary protein–protein interactions of the proteins coded by the differentially co-expressed genes, (2) interactions between differentially co-expressed genes’ and their upstream regulators, and (3) interactions between differentially co-expressed genes, microRNAs, and associated transcription factors; the networks were constructed and analyzed using Cytoscape 3.7.1 [77] and NetworkAnalyst 3.0 [78];The topological robustness analysis of the above networks (graphs) by the principle of topological attack on biological network; betweenness centrality identified nodes with the highest network-disrupting potential as possible targets for pharmacological interventions.

## 5. Conclusions

In our pilot study, we document that the airway epithelial transcriptome of eosinophilic vs. non-eosinophilic asthma patients, distinguished by BAL eosinophilia alone, differs regardless of an admixture of other inflammatory cells (e.g., neutrophils). The broad spectrum of analyses, specifically the extensive gene co-expression data, was intended to indicate a functional group of genes that might be mechanistically related to EA pathology and reveal common underlying regulatory processes, including receptor signaling, and the interference of miRNAs. Our study of the varied, real-life group of patients deviating from the pure EA phenotype did not reveal canonical markers of the T2 response (such as *CST1* or *CLCA1*). This suggests that such a heterogenous EA phenotype identified based on BAL eosinophilia and allowing for mixed granulocytic infiltration may only partially overlap with the pure T2 immunological signature. For that reason, we also included an extended literature search to describe important biological roles of the identified genes, including the inferred signaling pathways possibly involved in asthma and airway remodeling pathology.

Among the 32 differentially expressed genes, the expression of 11 correlated significantly with the measures of lung function and HRCT measures of airway remodeling, following their mechanistic roles. The protein products of these genes (in particular, of *GSN*, *NDUFAF8*, *KIR3DS1*, *KRTAP10.1*, *TBC1D12*, *RAB4B*, and *ATPIF1*) could be investigated as potential novel biomarkers and therapeutic targets against airway remodeling in EA.

Of the 32 differentially co-expressed genes, the products of *CCT7*, *EPS15*, *STRN4*, *SRPRB*, *GPI*, and *RAPH1* could be considered additional candidates for therapeutic interventions due to their mechanistic relevance for asthma and airway remodeling and their high connectivity in the related protein–protein interaction networks.

Inferences from the differential gene co-expression networks suggest that in EA, there is a preferential engagement of regulatory processes indicating the activation of the SMAD2/3, activator protein-1, E2F/Rb, Wnt/β-catenin, and androgen receptor signaling pathways, selected protein kinases TGFBR2, HIPK1, and IKBKE, and interference from microRNAs miR-135b110, miR-135a, miR-340, and miR-223. Furthermore, a considerable proportion of the differentially co-expressed genes matched patterns resulting from histone H3K9ac and H3K27me3 modifications, suggesting such epigenetic changes as being specific to EA.

However, future multimodal investigations are required to confirm the diagnostic and therapeutic utility of the presented findings.

## Figures and Tables

**Figure 1 ijms-24-03789-f001:**
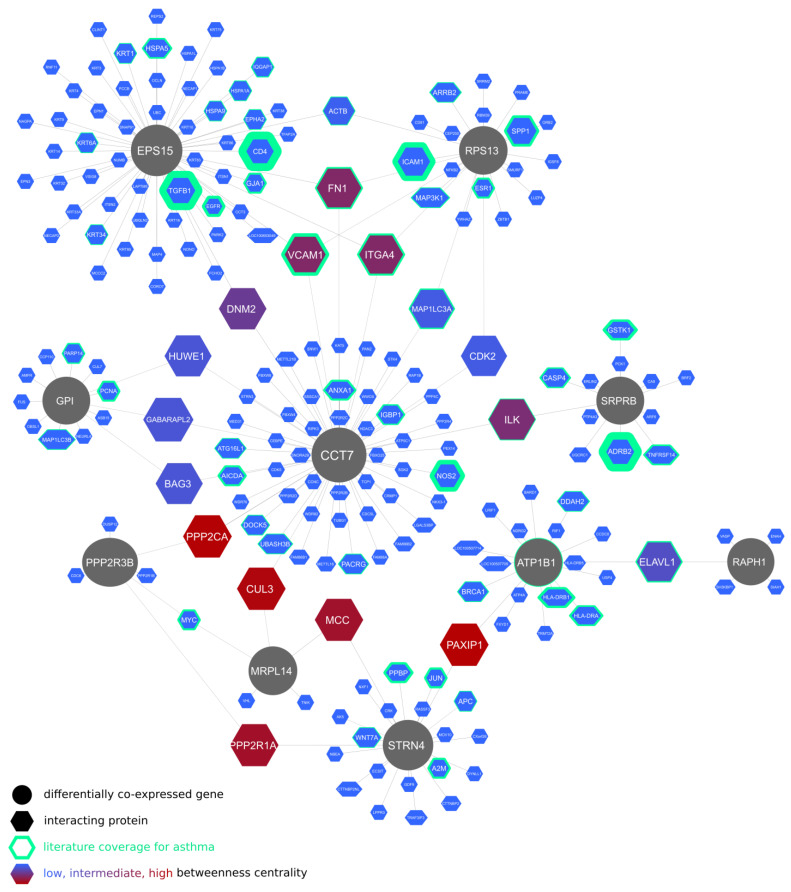
Lung-specific protein–protein interactions of genes differentially co-expressed between eosinophilic and non-eosinophilic asthma patients in bronchial brush biopsy samples.

**Figure 2 ijms-24-03789-f002:**
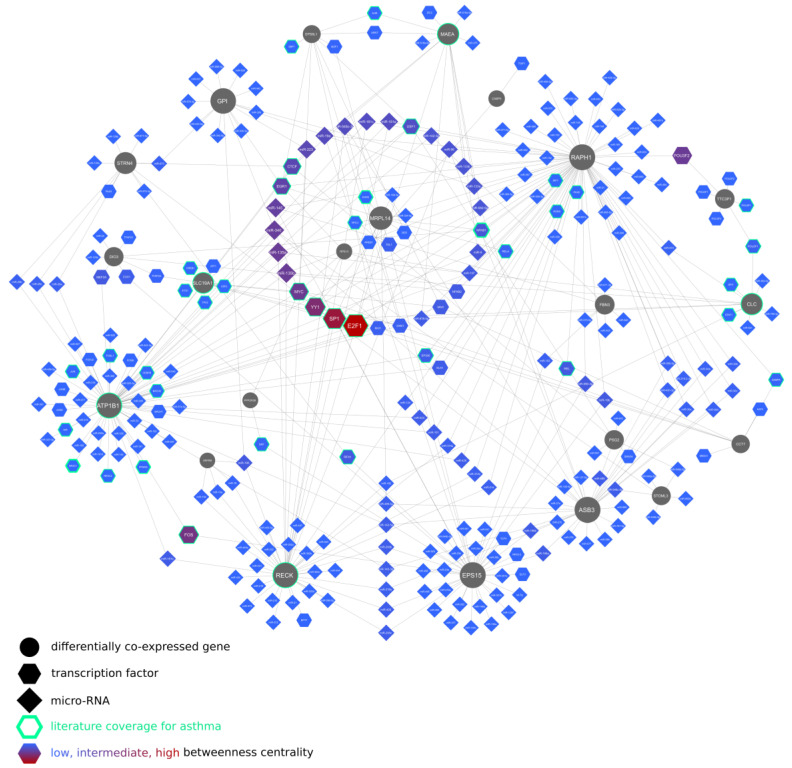
Regulatory network of miRNAs, transcription factors, and genes differentially co-expressed between eosinophilic and non-eosinophilic asthma patients in bronchial brush biopsy samples.

**Figure 3 ijms-24-03789-f003:**
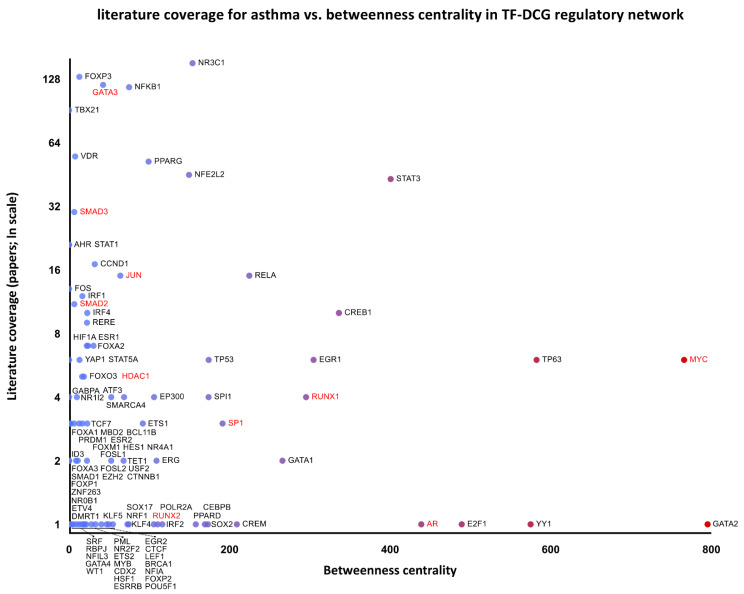
Literature coverage for asthma against betweenness centrality of the transcription factors–differentially co-expressed genes (TF-DCG) regulatory network. Transcription factors combined from networks are included in Appendix A. Genes labeled in red form the SMAD2/3 signaling pathway.

**Table 1 ijms-24-03789-t001:** Demographic and clinical characteristics of the patients.

Sample’s Clinical Features	EA	NEA	Difference *p*-Value *
Clinical features, confounding diseases			
Sample size	*n* = 20	*n* = 20	
Gender, *n* (%) female	13 (65%)	16 (80%)	0.240
Age, years ******	58 (16)	52 (23)	0.096
BMI, kg/m^2^ ******	25.4 (3.1)	30.4 (11.4)	**0.020**
Atopy, *n* (%)	11 (58%)	9 (53%)	0.515
Allergic rhinitis, *n* (%)	15 (88%)	17 (89%)	0.655
Gastroesophageal reflux disease, *n* (%)	8 (44%)	10 (58%)	0.305
Nicotinism, past, *n* (%)	13 (65%)	16 (80%)	0.563
Asthma severity			
ACT, points ******	14 (8)	16 (9)	0.227
Severe asthma (GINA class 4), *n* (%)	9 (45%)	7 (35%)	0.519
Asthma phenotype by different criteria			
BAL eosinophilia ≥ 1%	20 (100%)	none	n/a
BAL eosinophilia ≥ 2.5%	11 (55%)	none	n/a
BAL eosinophilia > 3%	8 (40%)	none	n/a
Blood eosinophilia ≥ 350/mcL	11 (55%)	4 (20%)	0.008
BAL neutrophilia > 40%	1 (5%)	none	n/a
BAL neutrophilia ≤ 40% and BAL eosinophilia < 1%	none	20 (100%)	n/a
Medications used			
Oral steroid, *n* (%)	5 (25%)	2 (10%)	0.237
Inhaled steroid, *n* (%)	15 (75%)	16 (80%)	0.962
LABA, *n* (%)	16 (80%)	15 (75%)	0.412
LAMA, *n* (%)	1 (5%)	4 (20%)	0.135
Montelukast, *n* (%)	2 (10%)	3 (15%)	0.677
Theophylline, *n* (%)	3 (15%)	2 (10%)	0.630
Acetylsalicylic acid/thienopyridines, *n* (%)	4 (20%)	8 (40%)	0.630
ACEI/ARB, *n* (%)	2 (10%)	6 (30%)	0.197
Beta-blockers, *n* (%)	3 (15%)	5 (25%)	0.110
Calcium channel blockers, *n* (%)	4 (20%)	7 (35%)	0.331
Diuretics, *n* (%)	4 (20%)	7 (35%)	0.331
Statins, *n* (%)	11 (55%)	9 (45%)	0.278
Proton pump inhibitors, *n* (%)	1 (5%)	4 (20%)	0.402

ACT—asthma control test, GINA—Global Initiative for Asthma, LABA—long-acting beta-agonists, LAMA—long-acting muscarinic antagonists, ACEI—angiotensin-II-converting enzyme inhibitors, ARB—angiotensin receptor blockers. Statistics: ***** Mann–Whitney U-test or Pearson’s chi-squared test when applicable, ****** median (interquartile range); differences of statistical significance at *p* < 0.05 are marked in **bold**; n/a—not applicable.

**Table 2 ijms-24-03789-t002:** Sample’s spirometry and high-resolution computed tomography proxies of airway remodeling.

Proxies of Airway Remodeling	EA	NEA	Difference *p*-Value *
Spirometry (with reversibility test) and body plethysmography			
FEV_1_, L	2.1 (0.79)	2.28 (1.26)	0.196
FEV_1_, % predicted	77.4 (40.2)	89.7 (31.1)	0.421
VC, L	3.33 (1.78)	3.36 (1.11)	0.292
FEV_1_/VC	62.86 (16.86)	67.8 (17.1)	0.267
∆FEV_1_, L	0.21 (0.23)	0.17 (0.22)	0.506
∆FEV_1_, %	0.118 (0.143)	0.07 (0.13)	0.419
TLC, L	6.65 (2.13)	5.63 (1.51)	0.419
TLC, % predicted	114.25 (18.5)	111.9 (22.5)	0.295
RV% TLC	30.63 (3.68)	30.45 (11.48)	0.913
High-resolution computed tomography (at RB1 bronchus)			
Wall thickness (mm, average)	1.216 (0.32)	1.078 (0.22)	**0.033**
Lumen diameter (mm, average)	2.719 (0.84)	2.344 (0.47)	0.077
Airway diameter (mm, average)	5.238 (1.58)	4.593 (0.68)	**0.044**
Wall thickness ratio	14.692 (2.2)	14.817 (1.94)	0.879
Wall area (mm^2^)	25.365 (12.81)	18.708 (6.56)	**0.033**
Wall area ratio	44.24 (4.99)	45.290 (4.98)	0.676

FEV_1_—forced expiratory volume in 1 s, VC—vital capacity, TLC—total lung capacity, RV—residual volume. ***** Mann–Whitney U-test; differences of statistical significance at *p* < 0.05 are marked in **bold**.

**Table 3 ijms-24-03789-t003:** Immune mediators in serum and bronchoalveolar lavage fluid.

Serum and BAL Studies	EA Group	NEA Group	Difference *p*-Value *
Serum cytokines, pg/mL (mean ± SD)			
IFN-γ	0.575 ± 1.406	0.405 ± 1.355	0.490
IL-	0.005 ± 0	0.005 ± 0	n/a
IL-5	0.005 ± 0	0.946 ± 4.210	0.797
IL-6	2.058 ± 2.059	1.366 ± 1.508	0.387
IL-10	1.260 ± 1.074	0.602 ± 0.807	**0.017**
IL-12p70	1.941 ± 3.777	1.255 ± 2.640	0.946
IL-17A	0.188 ± 0.579	0.171 ± 0.450	0.860
IL-23	15.81 ± 25.11	48.53 ± 98.36	0.323
Periostin	1.284 ± 3.038	0.329 ± 0.086	0.340
ADAM33 (ng/mL)	2.229 ± 1.612	1.462 ± 0.944	0.126
IgE (IU/mL)	161 (582)	32 (124)	**0.018**
BAL cytokines, pg/mL (mean ± SD)			
IFN-γ	0.005 ± 0	0.005 ± 0	n/a
IL-4	0.645 ± 1.832	0.379 ± 1.276	0.931
IL-5	0.342 ± 1.509	0.005 ± 0	0.342
IL-6	0.586 ± 0.583	0.832 ± 0.624	0.197
IL-10	0.005 ± 0	0.005 ± 0	n/a
IL-12p70	0.068 ± 0.021	0.063 ± 0.033	0.685
IL-17A	0.005 ± 0	0.005 ± 0	n/a
IL-23	0.245 ± 1.076	0.005 ± 0	0.342
Periostin	0.741 ± 0.210	0.911 ± 0.143	0.011
Blood cell count, cells/µL (mean ± SD)			
Neutrophils	3580 (2100)	3480 (1260)	0.370
Lymphocytes	2240 (1110)	1930 (730)	0.132
Eosinophils	480 (880)	140 (260)	**0.008**
Basophils	40 (30)	20 (20)	**0.012**
Monocytes	700 (370)	505 (110)	**0.020**
BAL cell %, median (range)			
Neutrophiles	4.0 (0.0–85.5)	3.0 (0.0–17.0)	0.211
Lymphocytes	8.0 (0.1–67.0)	4.0 (1.0–41.0)	0.052
Eosinophils	2.3 (1.0–42.0)	0.0 (0.0–0.5)	**<0.001**
Macrophages	80 (0.0–92.0)	92.8 (53.5–98.5)	**<0.001**
Cellularity, per ml	13,920 (253–42,250)	15,019 (6020–42,750)	0.157

***** Mann–Whitney U-test; differences of statistical significance at *p* < 0.05 are marked in **bold**. n/a—not available.

**Table 4 ijms-24-03789-t004:** Genes differentially expressed in bronchial brush biopsy samples of eosinophilic vs. non-eosinophilic asthma patients adjusted for age, sex, BMI, and oral corticosteroid use. Gene function data obtained from UniProtKB/Swiss-Prot database [15], unless stated otherwise.

Gene Name	Encoded Protein	log2FC	Adjusted *p*-Value	Gene Function and Relevance in Asthma
Genes up-regulated in eosinophilic asthma				
*CPLX2*	Complexin 2	0.077	0.00016	Regulates the formation of synaptic vesicles; likely plays a role in immunoglobulin secretion from plasmocytes.
*RTN4RL1*	Reticulon 4 Receptor Like 1	0.105	0.01339	Surface receptor found on vagal sensory neurons, likely involved in bronchial hyperreactivity.
*KRTAP10-1*	Keratin Associated Protein 10-1	0.070	0.01105	Forms a matrix of keratin intermediate filaments; also involved in nervous system development.
*KIR3DS1*	Killer Cell Immunoglobulin-Like Receptor, Three Ig Domains And Short Cytoplasmic Tail 1	0.088	0.01721	Natural killer cell immunoglobulin-like receptor for MHC class I molecules transducing activating signals; triggers degranulation and antiviral cytokine secretion upon interaction.
*AL137028.1*	IQ Motif And Sec7 Domain 3 Pseudogene 3	0.110	0.00348	A pseudogene of undetermined function.
*FBLN1*	Fibulin 1	0.075	0.01134	A structural protein stabilizing ECM proteins; regulates the biological availability of latent TGF-β [16].
*CDH23*	Cadherin Related 23	0.120	0.01721	Member of cadherin superfamily playing a role in cell–cell adhesion.
*SOX14*	SRY-Box Transcription Factor 14	0.055	0.01028	Transcription factor involved in angiogenesis, tissue injury, and wound healing.
*TMEFF1*	Transmembrane Protein With EGF Like And Two Follistatin Like Domains 1	0.081	0.00895	Found to be expressed upon stimulation with TGF-β as direct target gene of TGF-β2/Smad2/3; involved in airway remodeling.
*NDUFAF8*	NADH:Ubiquinone Oxidoreductase Complex Assembly Factor 8	0.075	0.00895	Involved in the assembly of mitochondrial NADH: ubiquinone oxidoreductase complex participating in reactive oxygen species production.
*ENTR1*	Endosome Associated Trafficking Regulator 1	0.106	0.0008	An endosomal protein that functions in actin cytoskeleton remodeling, protein trafficking, and secretion, cytokinesis, and apoptosis.
*CNN1*	Calponin 1	0.072	0.01928	Filament-associated protein implicated in the regulation of smooth muscle contraction.
*GSN*	Gelsolin	0.154	0.00096	Implicated in TGF-β-dependent smooth muscle actin synthesis in myofibroblasts and epithelial–mesenchymal transition [17].
*FMNL1*	Formin Like 1	0.065	0.00703	Plays a role in regulation of cell morphology and cytoskeletal organization, modifies macrophage motility.
*SUN5*	Sad1 and UNC84 Domain-Containing 5	0.062	0.01570	SUN family builds a bridge between the nucleoskeleton and the cytoskeleton, mediating nuclear dynamics during cell division.
*GYS2*	Glycogen Synthase 2	0.085	0.01267	Up-regulated upon signaling through PPARα, a major regulator of airway inflammation.
*SLC22A18AS*	Solute Carrier Family 22 Member 18 Antisense	0.102	0.00895	Antisense partner of an *SLC22A18* gene; its up-regulation may suppress *LC22A18*, an important tumor-suppressor gene, increasing cell proliferation.
*ANK3*	Ankyrin G	0.092	0.00895	The integral membrane protein that plays a role in cell motility, activation, proliferation, and contact.
*ATPIF1*	ATP Synthase Inhibitory Factor Subunit 1	0.071	0.01339	Its overexpression triggers reactive oxygen species production and promotes transcriptional activation of NFκB, resulting in a proliferative response involved in tissue remodeling.
*RAB4B*	Ras-Related GTP-Binding Protein 4b	0.090	0.00096	Plays a role in the regulation of vesicular trafficking.
Genes down-regulated in eosinophilic asthma				
*DNAJA1*	DnaJ Heat Shock Protein Family (Hsp40) Member A1	−0.075	0.01339	Down-regulation of *DNAJA1* was indicated as one of the biomarkers of isocyanate occupational asthma [18].
*GJB1*	Gap Junction Protein Beta 1	−0.040	0.03043	A gap junction channel protein; its absence inhibits TNF-α-induced extrinsic apoptosis pathway.
*TBC1D12*	TBC1 Domain Family Member 12	−0.088	0.00096	Negatively regulates neurite outgrowth; down-regulation in eosinophilic asthma may result in increased neurite sprouting in the airways, a feature of airway remodeling in asthma.
*GDPD1*	Glycerophosphodiester Phosphodiesterase Domain Containing 1	−0.064	0.00703	One of the proteins involved in glucocorticoid signaling pathways (PAHS-154Z); down-regulation may be associated with decreased sensitivity to glucocorticoids or prolonged glucocorticoid therapy.
*OR10H5*	Olfactory Receptor Family 10 Subfamily H Member 5	−0.067	0.01801	An olfactory receptor regulating cytoskeletal remodeling and ASM cell proliferation [19].
*DMRTC2*	DMRT Like Family C2	−0.049	0.01301	A transcription factor involved in H3K9 methylation pattern and TGF-β epithelial–mesenchymal transition in airway remodeling [20].
*SIN3A*	SIN3 Transcription Regulator Family Member A	−0.057	0.01268	A transcriptional repressor antagonizes the effects of MYC. Regulates the vast majority of the transcriptional response to hypoxia.
*HYMAI*	Hydatidiform Mole Associated And Imprinted	−0.055	0.01301	A non-protein coding gene of unknown function in airway pathology.
*UGT2A3*	UDP Glucuronosyltransferase Family 2 Member A3	−0.057	0.03012	One of the genes implicated in the metabolism of xenobiotics, including nicotine and tobacco carcinogens.
*CREB5*	CAMP Responsive Element Binding Protein 5	−0.038	0.01350	Recently identified as a common regulon in asthma exacerbations, expressed in macrophages and dendritic cells.
*ITM2B*	Integral Membrane Protein 2B	−0.100	0.00895	Associated with pediatric obesity-related asthma cases [21].
*CDC37P1*	Cell Division Cycle 37 *Pseudogene* 1	−0.057	0.00895	A pseudogene of undetermined function.
*C3orf23*	T Cell Activation Inhibitor, Mitochondrial	−0.041	0.02436	A mitochondrial protein implicated in apoptosis of T-cells.

**Table 5 ijms-24-03789-t005:** Correlations between expression levels of the differentially expressed genes, lung function parameters, and high-resolution computed tomography proxies of airway remodeling.

Proxy of Airway Remodeling	Differentially Expressed Gene	Pearson’s Correlation Coefficient	Correlation *p*-Value
Spirometry (with reversibility test) and body plethysmography			
FEV_1_, L	*RAB4B*	−0.64	0.001
	*FMNL1*	−0.51	0.002
	*AL137028.1*	−0.51	0.002
	*GSN*	−0.45	0.008
	*ANK3*	−0.40	0.018
	*SDCCAG3*	−0.39	0.024
	*NDUFAF8*	−0.36	0.036
	*KRTAP10.1*	−0.36	0.036
	*TBC1D12*	0.41	0.017
FEV_1_/VC	*GSN*	−0.53	0.001
	*AL137028.1*	−0.49	0.004
	*KIR3DS1*	−0.48	0.004
	*NDUFAF8*	−0.47	0.005
	*CDH23*	−0.44	0.010
	*KRTAP10.1*	−0.42	0.013
	*RTN4RL1*	−0.38	0.028
	*TBC1D12*	0.41	0.017
TLC, % predicted	*SIN3A*	−0.41	0.032
	*GSN*	0.42	0.028
	*SUN5*	0.44	0.018
	*FMNL1*	0.45	0.016
RV, %TLC	*DMRTC2*	−0.41	0.033
High-resolution computed tomography (at RB1 bronchus)			
Wall thickness (mm, average)	*ATPIF1*	0.49	0.006
Airway diameter (mm, average)	*ATPIF1*	0.48	0.007
Wall area (mm^2^)	*ATPIF1*	0.51	0.004
Wall area ratio	*RAB4B*	0.42	0.023
	*GSN*	0.39	0.034
	*NDUFAF8*	0.38	0.041

FEV_1_—forced expiratory volume in 1 s, VC—vital capacity, TLC—total lung capacity, RV—residual volume.

**Table 6 ijms-24-03789-t006:** Differentially co-expressed genes in bronchial brush biopsy samples of eosinophilic and non-eosinophilic asthma patients. Data obtained from UniProtKB/Swiss-Prot database [15], unless stated otherwise.

Gene Name	Encoded Protein	log2FC	Gene Function and Relevance in Asthma
Genes up-regulated in eosinophilic asthma			
*ATP1B1*	Sodium/potassium-transporting ATPase subunit beta-1	0.534	A ubiquitous protein regulating the location and function of sodium–potassium ATPase with possible involvement in epithelial sheathing, CMV and RSV infection, Th17 polarization, activation of inflammatory and airway smooth muscle cells, mucus secretion, and airway hyperreactivity [22].
*STRN4*	Striatin-4	0.481	A regulatory subunit of the STRIPAK complex involved in cell cycle control, cell adhesion, migration, epithelial integrity, and epithelial–mesenchymal transition; associated with asthma in an EWAS study [23].
*GPI*	Glucose-6-phosphate isomerase	0.349	An enzyme exhibiting the function of neurotrophic factor, a lymphokine inducing immunoglobulin secretion and an angiogenic factor, involved in secretion of TNF-α and IL-1b.
*ANKRD26P1*	Ankyrin repeat domain 26 pseudogene 1	0.259	A pseudogene of little-known function.
*RAPH1*	Ras association (RalGDS/AF-6) and pleckstrin homology domains	0.245	An adaptor protein regulating actin dynamics.
*RP3-473B4.3*	(lincRNA; not named yet)	0.145	No direct association with asthma, inflammation, or airway remodeling is known.
*CLC*	Charcot–Leyden crystal protein	0.124	An atypical galectin with an activity of lysophospholipase; taking part in vesicular transport of eosinophilic granule ribonucleases; ubiquitous in sputum of eosinophilic asthma patients.
*RADX*	RPA1 related single stranded DNA binding protein, X-linked	0.107	One of the genes up-regulated upon transition from pericytes into mesenchymal cells; may indicate an epithelial–mesenchymal transition in airway remodeling.
*RECK*	Reversion inducing cysteine rich protein with kazal motifs	0.084	A membrane-anchored protein negatively regulating a matrix metalloproteinase-9, a major metalloproteinase involved in asthma pathology [24].
*DGLUCY*	D-glutamate cyclase	0.076	A mitochondrial enzyme known to up-regulate E-cadherin and repress signaling through ERK, possibly limiting epithelial–mesenchymal transition.
*SLC19A1*	Solute carrier family 19 member 1	0.068	Takes part in folate, homocysteine, nitric oxide, and reactive oxygen species metabolism and DNA methylation; possibly linked with asthma through effects of folate deficiency resulting in aggravated asthma symptoms, increased CD4/CD8 T-cell ratio, extracellular matrix remodeling, and signalling through β-catening/Wnt pathway implied in airway remodeling. [25]
*TTC3P1*	Tetratricopeptide repeat domain 3 pseudogene 1	0.060	As a pseudogene, it may interfere with *TTC3*, a gene for E3 ubiquitin-protein ligase.
*RP11-321E2.2*	(lincRNA; not named yet)	0.052	No direct association with asthma, inflammation, or airway remodeling is known.
*MAEA*	macrophage erythroblast attacher	0.049	A ubiquitin ligase up-regulated upon exposure to cigarette smoke extract, RSV and MMP-9 [26]; an EWAS study found an association between trans-CpG site in *MAEA* and *IL1RL1* [27].
*FBN3*	Fibrillin-3	0.035	A component of extracellular calcium-binding myofibrils occurring with elastin- or elastin-free bundles important for ECM integrity; may be involved in the regulation of TGF-β signalling through association with latent TFG-β-binding proteins.
*PSG2*	Pregnancy-specific beta-1-glycoprotein 2	0.031	Stimulates transcription of *FOXP3* in mononuclear and CD4+ T-cells providing a signal for T-reg and Th17 differentiation [28].
*PPP2R3B*	Protein phosphatase 2 regulatory subunit B beta	0.024	A positive regulator of cell cycle progression.
*GOLGA2P3Y*	Golgin A2 Pseudogene 3	0.023	No direct association with asthma, inflammation, or airway remodeling is known.
*STOML3*	Stomatin like 3	0.025	Plays a role in mechanotransduction through interaction with PIEZO1 with possible implications for airway hyperreactivity, cell adhesion, and migratory capacity of epithelial cells through inactivation of β_1_ integrin affinity [29].
*DIO3*	Deiodinase iodothyronine type III	0.019	An enzyme inactivating thyroid hormones; induced by TGF-β and oxidative stress.
*FADS6*	Fatty acid desaturase 6	0.017	A peroxisomal enzyme taking part in polyunsaturated fatty acids biosynthesis; contributes to arachidonic acid synthesis; regulated by miR-331-3p, a post-transcriptional regulator associated with lung function in asthma [30].
*EPS8L1*	EPS8 like 1	0.012	A protein related to Eps8 involved in actin remodeling; plays a role in T-cell receptor binding, membrane ruffling, and remodeling of the actin cytoskeleton by CDC42 relevant for cell migration.
*RPS13*	Ribosomal protein S13	0.013	A housekeeping gene forming the TNF-α/NFκB signalling complex of established role in asthma.
*ASB3*	Ankyrin Repeat And SOCS Box Containing 3	0.008	Part of the ASB gene family involved in Erk1/2 and PI3K/Akt signal transduction pathways, implied in airway remodeling.
*MRPL14*	Mitochondrial ribosomal protein L14	0.009	Controlled by MYC transcription factor; related to asthma in a GWAS study [31].
Genes down-regulated in eosinophilic asthma			
*EPS15*	Epidermal growth factor receptor pathway substrate 15	−0.004	A protein involved in internalization of tyrosine kinase receptors (including TGF-β), integrins (including integrin β_1_ and E-cadherin), and receptors relevant for bronchoconstriction (β_2_-adrenergic and M_3_-muscarinic) [32].
*RHOBTB2*	Rho related BTB domain containing 2	−0.010	Required for expression of CXCL14, a chemoattractant controlling dendritic cell activation, leukocyte migration, angiogenesis, and an autocrine growth factor for fibroblasts; differentially expressed after bronchial thermoplasty.
*SRCRB4D*	Scavenger Receptor Cysteine Rich Family Member With 4 Domains	−0.016	Possibly associated with asthma as one of the target genes of BACH1 involved in response to oxidative stress and associated with asthma in an EWAS study [23].
*OR52I1*	Olfactory receptor family 52 subfamily I member 1	−0.098	An olfactory receptor possibly involved in airway hyperreactivity.
*CCT7*	Chaperonin containing TCP1 subunit 7	−0.141	A molecular chaperon up-regulated in fibrotic wound healing and essential for accumulation of α-smooth muscle actin in fibroblasts and differentiation to myofibroblasts [33]; recently found to coimmunoprecipitate with thromboxane A_2_ receptor and β_2_-adrenergic receptor [34]; its depletion resulted in reduced expression of both receptors involved in asthma pathogenesis.
*SRPRB*	Signal recognition particle receptor beta subunit	−0.162	Involved in intracellular trafficking of proteins.
*CABP5*	Calcium-binding protein 5	−0.602	Reported in T-cells and differentially expressed in allergic asthma patients [35].

## Data Availability

The data presented in this study are available on request from the corresponding author.

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
