# Peer review of "Co-Expression Analysis of Airway Epithelial Transcriptome in Asthma Patients with Eosinophilic vs. Non-Eosinophilic Airway Infiltration"

_ijms, 2023, doi:10.3390/ijms24043789_

Round 1

Reviewer 1 Report

This study sought molecular mechanisms involved in the pathogenesis of eosinophilic asthma (EA) by analysis of bronchial epithelium transcriptomes in concert with systems biology approach including measurements of inflammatory mediators (in BAL and blood), lung function, and airway remodeling index, etc. Authors also made multiway correlations between indicators acquired from those measurements in several ways. Tremendous experimentations led to production of copious amounts of results along with citations of extraordinary numbers (155) of publications for a regular article. Some positive and significant results are well described in Discussion section, especially including better utility of EA and NEA for classification of heterogenous asthma, BAL eosinophils rather than blood eosinophils, and T2 cytokine profile, etc.

Major comments:

It is difficult for me to find key findings that would bring new insights to EA pathology. Authors intended to discern how EA differs from NEA at the molecular level, as they performed transcriptomic analysis as the main experimental tool, but no biomarker or predictor cannot be actually specified found for this purpose from this study. Therefore, at least some of the suggested genes should be validated for their involvement in EA vs NEA pathologies.

Minor comments:

Several misspells are found: lines 64, 330, and many others.  

Citation of many articles in references are not consistent or appropriate.   

Author Response

Response to the Reviewer #1 regarding manuscript titled Co-expression analysis of airway epithelial transcriptome in asthma patients with eosinophilic vs. non-eosinophilic airway infiltration.

Comments and suggestions from the Reviewer #1:

This study sought molecular mechanisms involved in the pathogenesis of eosinophilic asthma (EA) by analysis of bronchial epithelium transcriptomes in concert with systems biology approach including measurements of inflammatory mediators (in BAL and blood), lung function, and airway remodeling index, etc. Authors also made multiway correlations between indicators acquired from those measurements in several ways. Tremendous experimentations led to production of copious amounts of results along with citations of extraordinary numbers (155) of publications for a regular article. Some positive and significant results are well described in Discussion section, especially including better utility of EA and NEA for classification of heterogenous asthma, BAL eosinophils rather than blood eosinophils, and T2 cytokine profile, etc.

Major comments:

It is difficult for me to find key findings that would bring new insights to EA pathology. Authors intended to discern how EA differs from NEA at the molecular level, as they performed transcriptomic analysis as the main experimental tool, but no biomarker or predictor cannot be actually specified found for this purpose from this study. Therefore, at least some of the suggested genes should be validated for their involvement in EA vs NEA pathologies.

Answer to the Reviewer:

The authors would like to thank the Reviewer for the comprehensive assessment of the work. We do agree with this comment. Indeed, presented results, described in the formal order of consecutively performed analyses, are challenging to follow. Therefore, we removed potentially redundant data and additionally clarified the critical aspects of the study in the Conclusions section in an updated version of the manuscript (page 19, lines 458-491). For example, we detailed the purpose of co-expression data analyses and indicated genes that might be future study targets through further genomic and proteomic investigations. We believe that the central message of the article is now more transparent.

We must admit, however, that our study should be regarded as a pilot project. The broad spectrum of analyses, specifically the extensive co-expression data, was intended to indicate genes or rather gene sets (functional groups of genes) that might be mechanistically related to eosinophilic asthma (EA) pathology, including regulatory processes, receptor signaling, and miRNAs. Importantly, analysis of the varied groups of patients (in fact, real-life cohorts with not always pure EA phenotype) did not reveal canonical markers of T2-response (such as CST1 or CLCA1). It suggests that EA phenotype identified based on BAL fluid is quite heterogeneous in terms of gene expression in the bronchial brush, and may only partially overlaps with pure T2 immunological signature. For that reason, we also included an extended search of the literature to describe important biological roles of identified genes, including inferred signaling pathways possibly involved in asthma and airway remodeling pathology. Hence, our work has this specific review appearance with more than 100 references cited. In the future, we plan to validate selected genes, e.g., those related to airway remodeling measures, in more detailed genetic and protein studies.

We clarified that issue in the revised version of the manuscript in the Conclusions subsection.

Minor comments:

  • Several misspells are found: lines 64, 330, and many others.

Answer to the Reviewer: We have corrected the spelling errors

  • Citation of many articles in references are not consistent or appropriate.

Answer to the Reviewer: We have corrected the citation list. Some references have been removed, and all remaining have been checked for correctness.

Sincerely,

On behalf of the co-authors,

Stanislawa Bazan-Socha, M.D., Ph.D.

Reviewer 2 Report

The authors of this very interesting and useful paper investigate relations between inflammation, bronchial gene expression, and functional and structural measures of airway remodeling in eosinophilic and non-eosinophilic asthma. The results are presented and interpreted appropriately and support the conclusions. The study is correctly designed and technically sound, using methods, tools, software, and reagents described with sufficient details to allow other researchers to reproduce the results. The paper is wonderfully written, with appropriate and understandable use of the English language. This work, which advances the current knowledge, should interest a great number of people, as asthma is an extremely common chronic disease, inducing a great burden on patients and healthcare systems. The paper can be accepted without any further changes.

Author Response

Response to the Reviewer #2 regarding manuscript titled Co-expression analysis of airway epithelial transcriptome in asthma patients with eosinophilic vs. non-eosinophilic airway infiltration.

Comments and suggestions from the Reviewer #2:

The authors of this very interesting and useful paper investigate relations between inflammation, bronchial gene expression, and functional and structural measures of airway remodeling in eosinophilic and non-eosinophilic asthma. The results are presented and interpreted appropriately and support the conclusions. The study is correctly designed and technically sound, using methods, tools, software, and reagents described with sufficient details to allow other researchers to reproduce the results. The paper is wonderfully written, with appropriate and understandable use of the English language. This work, which advances the current knowledge, should interest a great number of people, as asthma is an extremely common chronic disease, inducing a great burden on patients and healthcare systems. The paper can be accepted without any further changes.

Answer to the Reviewer: The authors would like to thank the Reviewer for the positive comments and appreciation of our work. 

Sincerely,

On behalf of the co-authors,

Stanislawa Bazan-Socha, M.D., Ph.D.

Round 2

Reviewer 1 Report

Many corrections were made to stand out improve the significance of this manuscript and improve readers' understanding. 

Author Response

Response to the Reviewer #1 regarding manuscript titled Co-expression analysis of airway epithelial transcriptome in asthma patients with eosinophilic vs. non-eosinophilic airway infiltration (following major revision).

Comments and suggestions from the Reviewer #1:

Many corrections were made to stand out improve the significance of this manuscript and improve readers' understanding.

Answer to the Reviewer:

The authors wish to express grattitute for the thorough assessment and appreciation of our work and the remarks which allowed us to make the necessary revisions.

Sincerely,

On behalf of the co-authors,

Stanislawa Bazan-Socha, M.D., Ph.D.
